# Bone Fracture Enhanced Blood-Brain Barrier Breakdown in the Hippocampus and White Matter Damage of Stroke Mice

**DOI:** 10.3390/ijms21228481

**Published:** 2020-11-11

**Authors:** Jinhao Huang, Haiyan Lyu, Kang Huo, Leandro B. Do Prado, Chaoliang Tang, Zhanqiang Wang, Qifeng Li, Julia Wong, Hua Su

**Affiliations:** 1Department of Anesthesia and Perioperative Care, University of California, San Francisco, CA 94143, USA; jinhao_huang@163.com (J.H.); luhaiyan198109@163.com (H.L.); huokang8204@xjtu.edu.cn (K.H.); Leandro.BarbosaDoPrado@ucsf.edu (L.B.D.P.); tangruoxiao2012@hotmail.com (C.T.); wangzhanqiang11@vip.163.com (Z.W.); Qifeng.Li@ucsf.edu (Q.L.); juywong@ucdavis.edu (J.W.); 2Center for Cerebrovascular Research, University of California, San Francisco, CA 94143, USA

**Keywords:** stroke, bone fracture, blood brain barrier breakdown, white matter damage, hippocampus

## Abstract

Background: Tibia fracture (BF) before stroke shortly causes long-term post-stroke memory dysfunction in mice. The mechanism is unclear. We hypothesize that BF enhances neuroinflammation and blood brain barrier (BBB) breakdown in the hippocampus and white matter (WM) damage. Methods: Mice were assigned to groups: BF, stroke, BF+stroke (BF 6 h before stroke) and sham. BBB integrity was analyzed 3 days after the surgeries and WM injury was analyzed 3 days and 8 weeks after the surgeries. Results: Stroke and BF+stroke groups had more activated microglia/macrophages and lower levels of claudin-5 in the ipsilateral hippocampi than the BF group. BF+stroke group had the highest number microglia/macrophages and the lowest level of claudin-5 among all groups and had fewer pericytes than BF group. Stroke and BF+stroke groups had smaller WM areas in the ipsilateral basal ganglia than the sham group 8 weeks after the injuries. The BF+stroke group also had smaller WM areas in the ipsilateral than sham and BF groups 3 days after the injuries and in the contralateral basal ganglia than stroke and BF groups 8 weeks after the injuries. Conclusions: BF exacerbates neuroinflammation and BBB leakage in the hippocampus and WM damage in basal ganglia, which could contribute to the long-lasting memory dysfunction in BF+stroke mice.

## 1. Introduction

Stroke is one of the major causes of death in the US (www.strokecenter.org/patients/about-stroke/stroke-statistics/). Despite the advance of medical research and the development of new technologies, the intravascular thrombectomy and tissue plasminogen activator (tPA) treatments are the limited therapeutic options available for the management of acute ischemic stroke patients [1,2]. More than 140,000 stroke patients die each year. Stroke patients have an increased risk of bone fracture [3,4,5,6,7]. Compared to the general population, the risk of bone fracture in stroke patients is about two- to four-fold higher. About 5% of stroke patients will suffer from a bone fracture [3,5,6,7,8]. After hip surgery, approximately 4% of patients will have a stroke within a year. Hip fracture doubles the risk of stroke [9,10]. Stroke patients with bone fractures have a higher mortality than those without, especially in older patients [11,12]. As the aging population increases, the number of patients with stroke and bone fracture will increase. We showed in animal models that bone fracture before or after ischemic stroke caused more neuronal damage and sensorimotor dysfunction [13,14,15,16]. These data suggest that bone fracture has a negative impact on stroke recovery.

Both stroke and big bone fracture increase the risk of dementia [17,18]. Dementia and stroke often occur together; the combination of these injuries significantly increases the mortality and the cost of care [19,20,21,22]. Stroke survivors are more likely to develop cognitive dysfunction than the general population [22,23,24,25]. Dementia in post-stroke patients may be caused by an associated comorbidity; however, even after controlling for dementia risk factors, such as hypertension, hypercholesterolemia, diabetes, and age, the occurrence of a single stroke increases the risk of developing new-onset dementia by at least two-fold [24,26].

Peripheral trauma, including bone fracture, can cause neuroinflammation and cognitive impairment [27,28]. For most patients, post-trauma or post-surgery neuroinflammation and cognitive decline resolve promptly with no sequelae. However, for patients with some risk factors, such as metabolic syndrome, Alzheimer’s disease, and aging, neuroinflammation can persist and cause persistent and even permanent cognitive dysfunction [29]. Cognitive dysfunction increases the risk of mortality [30]. Post-bone fracture cognitive decline is more severe in the elderly than in younger patients. About 10% of aged patients develop persistent cognitive decline after orthopedic surgeries. In young mice, bone fracture-caused memory dysfunction is temporary, shorter than 1 week [27,31]. We found that young mice developed long-lasting (≥ 8 weeks) spatial memory dysfunction when bone fracture preceded ischemic stroke by 6 h, which was associated with an accumulation of CX3C chemokine receptor 1^+^ (Cx3cr1^+^) cells in the hippocampal stratum lacunosum moleculare (SLM), stratum oriens and alveus [32]. The reasons why Cx3cr1^+^ cells accumulated in these regions and their association with long-term spatial memory dysfunction are currently unclear.

The blood-brain-barrier (BBB) contains tight-junction proteins, endothelial cells, and end-feet of astrocyte and pericytes [33,34]. BBB impairment is one of the representative pathological phenotypes of ischemic stroke. BBB impairment can lead to the extravasation of blood content into the brain parenchyma, enhancing local inflammatory injury [35]. Alternatively, inflammation can enhance the damage of the BBB. It has been shown that the infiltration of peripheral macrophages in the hippocampus leads to cognitive dysfunction [31,36]. We have detected the accumulation of microglia/macrophages in SLM of mice subjected to bone fracture and ischemic stroke, which was associated with a long-lasting memory dysfunction [32]. The connection of accumulation of microglia/macrophages in the hippocampi and memory dysfunction is not clear. In this paper, we showed that accumulation of microglia/macrophages correlated with an exacerbation of BBB leakage in the hippocampus and WM damage in basal ganglia in mice subjected to tibia fracture (BF) 6 h before ischemic stroke (BF+stroke). All of these could be the contributors of the long-lasting memory dysfunction observed in BF+stroke mice.

## 2. Results

### 2.1. Stroke and BF+Stroke Mice Have Increased Number of CD68^+^ Cells in the Hippocampal CA1 Region Ipsilateral to the Stroke Injury

To analyze if mice subjected to BF and ischemic stroke have more active microglia/macrophages in the hippocampi, we performed BF, permanent distal middle cerebral artery occlusion (pMCAO, ischemic stroke model), and BF 6 h before pMCAO in 8-weeks old C57 mice. Sham operated mice were used as control. The active microglia/macrophages in the hippocampi were analyzed 3 days after the surgeries by immunostaining using an anti-CD68 antibody (Figure 1). The numbers of CD68^+^ cells in ipsilateral hippocampal CA1 regions among groups were different (*p* < 0.001, one way-ANOVA). Multiple comparisons with Tukey’s post-hoc correction indicated that compared to the sham group (290 ± 17.6 cells/mm^2^), the stroke (381 ± 26.1, *p* = 0.007) and BF+stroke (437 ± 45.3, *p* < 0.001) groups had more CD68^+^ cells. The stroke (*p* = 0.004) and BF+stroke (*p* < 0.001) mice had more CD68^+^ cells than BF mice (310 ± 21.2). The BF+stroke mice also had more CD68^+^ cells than stroke mice (*p* = 0.035, Figure 2). Therefore, BF exacerbated neuroinflammation in the hippocampal CA1 region ipsilateral to the stroke injury.

### 2.2. BF Exacerbates BBB Breakdown in the Hippocampus

To test whether the BF exacerbates the BBB breakdown in the hippocampus, we analyzed the expression of tight junction protein, claudin-5, and vascular pericyte coverage in the brain sections collected 3 days after surgeries (Figure 1). CD31 was used as a maker for endothelial cells and CD13 was used as a marker for pericyte. The level of claudin-5 in capillaries and vascular pericyte coverage were expressed by the ratios of claudin-5 protein/CD31 and CD13/CD31.

One-way ANOVA analysis showed that the expression of claudin-5 was significantly different among groups (*p* < 0.001). Multiple compressions and Tukey’s post hoc analysis showed that compared to the sham group (0.45 ± 0.03), the stroke group (0.33 ± 0.03, *p* = 0.002) and BF+stroke group (0.25 ± 0.04, *p* < 0.001) had lower levels of claudin-5 in hippocampi ipsilateral to the stroke injury. The BF+stroke group (0.33 ± 0.02) also expressed a lower level of claudin-5 on the contralateral side than sham group (0.45 ± 0.01, *p* < 0.001) and BF group ((0.43 ± 0.02, *p* = 0.004). Compared to the BF mice (0.44 ± 0.02), the stroke (*p* = 0.008) and BF+stroke (*p* = 0.003) mice expressed lower levels of claudin-5 on the hippocampi ipsilateral side to stroke lesion. The stroke mice (*p* = 0.003) and the BF+stroke mice (*p* = 0.04) expressed lower levels of claudin-5 on the ipsilateral sides than contralateral sides (stroke: 0.42 ± 0.02 and BF+stroke: 0.33 ± 0.02) of hippocampi. The BF+stroke group also expressed a lower level of claudin-5 than the stroke mice on the ipsilateral side of stroke (*p* = 0.04, Figure 3). Therefore, BF enhanced the reduction of claudin-5 expression in the hippocampi of stroke mice.

One-way ANOVA analysis showed that the pericyte coverages were different among groups (*p* < 0.001). Multiple comparisons and Tukey’s post hoc analysis showed that the BF+stroke group (0.63 ± 0.03) has a lower pericyte coverage than the sham (0.79 ± 0.04, *p* < 0.001) and the BF groups (0.78 ± 0.11, *p* = 0.002) on the ipsilateral side of hippocampi (Figure 4).

Together, these data indicate that BF reduces tight junction proteins and vascular pericyte coverage in the hippocampi of mice subjected to ischemic stroke injury, leading to the impairment of the BBB integrity.

### 2.3. BF Exacerbated WM Damage in the Basal Ganglia of Stroke Brain

The WM area was analyzed by immunohistological staining of MBP 3 days and 8 weeks post-injuries (Figure 1). In mouse brain, the WMs are mostly recognizable in the corpus callosum and basal ganglia. Because the corpus callosum ipsilateral to the stroke lesion is often damaged, it may influence the accuracy of the measurement. Therefore, we only measured the WM area in the basal ganglia. The sizes of BMP positive areas were different among groups at both time points (*p* < 0.001, one-way ANOVA). Multiple comparisons and Tukey’s post hoc analysis showed that the MBP positive area on the ipsilateral side of stroke of the BF+stroke mice (0.09 ± 0.03 MPB positive area/total basal ganglia area) was smaller than the sham (0.14 ± 0.02) and BF groups (0.14 ± 0.03) at 3 days (*p* = 0.04, Figure 5A,B) after the injuries. At 8 weeks post injuries, the MBP positive areas of the BF mice (Ips: 0.09 ± 0.02; Con: 0.10 ± 0.03) were smaller than the sham group (Ips: 0.14 ± 0.02; Con: 0.14 ± 0.02) on both ipsilateral (*p* < 0.001) and contralateral sides (*p* = 0.003). The MBP positive area of the stroke (Ips: 0.04 ± 0.01; Con: 0.08 ± 0.01) and BF+stroke (Ips: 0.05 ± 0.01; Con: 0.06 ± 0.01) groups were also smaller than the sham group on both sides of the basal ganglia (*p* < 0.001). Compared to the BF group, the MBP positive area of the BF+stroke group was smaller on both the contralateral (*p* = 0.02) and ipsilateral sides (*p* = 0.008) of stroke. The Stroke group also had a smaller MBP positive area than the BF group on the ipsilateral side (*p* = 0.04, Figure 5A,C). Therefore, BF not only caused WM damage, but also exacerbated WM damage in the stroke brain.

## 3. Discussion

We showed in this study that in mice, BF shortly preceded ischemic stroke enhanced neuroinflammation in the hippocampus. There were more CD68^+^ cells in the hippocampal CA1 region of mice that had both BF and ischemic stroke injuries than mice that had BF or stroke alone. Mice subjected to BF and stroke also had lower levels of tight junction protein, claudin-5, and fewer vascular pericytes in the hippocampi than mice subject to single injury, suggesting that BF exacerbates BBB breakdown. Furthermore, BF reduced WM in the basal ganglia and enhanced WM damage in the basal ganglia of the stroke mice. All of this could contribute to the previously observed long-lasting memory dysfunction of BF+stroke mice [32].

It has been reported that in young healthy mice, BF increases neuroinflammation in the hippocampus, which is accompanied with a short-term (<7 days) memory dysfunction [31,37]. Augmentation of inflammation in the acute stage of stroke has a negative impact on the stroke outcome [38,39,40]. Our team reported previously that BF occurring 6 h or 24 h before or 24 h after stroke exacerbated neuroinflammation, brain edema and neuronal damage in the infarct and peri-infarct regions, and enhanced sensorimotor dysfunction of the stroke mice [13,14,15,16,41]. Our team had also shown that BF occurring 6 h before stroke led to long-lasting memory impairment (>8 weeks) of young mice [32]. Clinically, patients with stroke plus BF need more care and have poorer recovery than those with stroke alone [42]. The mechanism of how BF causes long-lasting memory dysfunction of stroke subjects has not been studied. A recent study showed that microglia play an important role in forgetting memory through eliminating synapses [43]. Future studies are needed to analyze if an increased number of activated microglia/macrophages in the hippocampus can lead to excessive synapses elimination, which in turn would cause memory dysfunction.

BBB breakdown causes edema, which enhances neuronal damage in the stroke area. The tight junction proteins are important components of the BBB. Reduction of tight junction protein expression impairs BBB integrity. Claudin-5, a major tight junction protein, seals the “gap” between endothelial cells junctions [44]. The pericytes play an important role in BBB, which not only form a supporting system (scaffold) for endothelial cells but also send paracrine signals by direct contact to the endothelium [45,46]. We have shown in our previous study that BF occurring 24 h after ischemic stroke exacerbated the reduction of cludin-5 protein in the peri-infarct region and enhanced brain edema [16]. It is not clear if BF occurring before or after stroke also exacerbate BBB breakdown in the hippocampi of stroke mice. In this study, we showed that the BF shortly before stroke significantly decreased the level of claudin-5 and the vascular pericyte-coverage in the hippocampi of the stroke mice, indicating that BF can exacerbate BBB breakdown in the hippocampi of stroke mice as well.

WM occupies half of the brain in humans. WM is important in the transmission of electric signals between neurons [47]. Approximately 64–86% stroke patients have WM injury, which correlates with a long-term sensorimotor and cognitive impairment [48,49]. We showed in this study that the WM was decreased in the basal ganglia of BF, stroke, and BF+stroke mice 8 weeks after the injuries. The reduction of WM was more severe in BF+stroke mice than in mice with BF or stroke alone. These data suggest that reduced WM volume caused by BF is one of the underlying mechanisms of long-term cognitive dysfunction in BF+stroke mice. More studies are needed to determine how BF leads to WM injury in the basal ganglia.

How a system injury interacts with brain and increases neuroinflammation causing brain damage has not been fully understood. A previous study showed that aseptic trauma activates the innate immune response through releasing high mobility group box 1 (HMGB1) from damaged tissues [50]. Through the interaction with pattern recognition receptors on circulating bone marrow-derived monocytes [50], HMGB1 induces the release of pro-inflammatory cytokines through transcriptional upregulation of NF-κB [51]. Peripheral inflammation disrupts BBB, allowing CCR2^+^ bone marrow derived cells entering the hippocampus and extravasation of plasminogen and albumin, which ultimately activates microglia and astrocytes [36,52,53,54,55]. Increased BBB permeability can increase the deposition of neurotoxins in the brain, leading to WM damage and cognitive dysfunction [56,57]. Together, these factors generate a neuroinflammatory response in the brain [58]. Our previous study showed that HMGB1 neutralization antibody treatment reduced neuronal injury in the peri-infarct regions of mice with stroke alone or stroke plus BF [13].

The limitations of this study are that we did not measure HMBG1 level nor did we test if the HMGB1 neutralization antibody can alleviate post-stroke memory dysfunction in BF+stroke mice. The other limitations include: (1) we did not test what type/size proteins had leaked out from blood vessels in the hippocampus; (2) we did not analyze whether the neuroinflammation caused damage of neural fibers and altered the number of synapses. The pathways involved in the negative impact of BF on post-stroke memory need to be exploited in future studies.

In summary, in this study, we showed that BF exacerbated BBB breakdown and neuroinflammation in the hippocampus, and enhanced WM damage in the basal ganglia of stroke mice. All of these could contribute to the long-term memory dysfunction observed in BF+stroke mice.

## 4. Materials and Methods

### 4.1. Animals

Animal experimental procedures were approved by the Institutional Animal Care and Use Committee (IACUC) of the University of California, San Francisco (The protocol is reviewed annually. Current approval number is AN185803, which covers 3 September 2020 to 3 September 2021), and were conformed to the guidelines of National Institutes of Health for the Care and Use of Laboratory Animals.

C57BL/6 male mice (8-weeks old) were purchased from the Jackson Laboratory (Bar Harbor, MA, USA) and housed in sawdust-lined cages (<5 mice/cage) with standard rodent food and water ad libitum in an air-conditioned environment with 12-h light/dark cycles.

The mice were assigned to four groups randomly: (1) BF; (2) stroke (pMCAO); (3) BF+stroke (pMCAO 6 h after BF); and (4) Sham (Sham BF and sham pMCAO procedures). Brain samples were collected 3 days or 8 weeks after the surgeries.

For all surgery procedures, 2% isoflurane inhalation was used to anesthetize mice. Buprenorphine (0.1 mg/kg) was given as analgesia before surgeries, 6 h after surgeries and as needed. Lidocaine was applied to the wounding area to reduce painful irritation. All mice were put in a temperature-controlled electric blanket to keep the rectal temperature around 37 ± 0.5 °C during the surgeries.

### 4.2. BF Surgery

BF was performed as described in our previous papers [14,15,16,41,59]. In brief, after the mice were anesthetized, an incision was made from the knee to the midshaft of the tibia. A hole (0.5 mm) was drilled in the proximal tibia just beneath and medial to the patellar tendon with a 25-gauge needle. A 0.38 mm stainless steel rod was inserted in the intramedullary canal until resistance was felt. An osteotomy was performed on the tibia with scissors at the junction of the middle and distal third of the bone. The wound was closed. The mice were allowed to recover from anesthesia spontaneously in warmed cages. BF sham was performed by hind limb hair shaving. The sham mice had received the same amount and duration of anesthesia and analgesia as those subjected to BF.

### 4.3. pMCAO Procedure

Six hours after BF, the mice were subjected to pMCAO, a procedure that has been described in our previously published papers [14,15,16,41,59]. Briefly, after the mouse was anesthetized, a skin incision (1-cm) was made between the left tragus and orbit. A 2 mm^2^ craniotomy was then done to expose the middle cerebral artery (MCA). The MCA just proximal to the pyriform branch was permanently occluded by electrocoagulation. The surface cerebral blood flow was monitored by a laser Doppler flow-meter (Vasamedics, Little Canada, MN, USA) to determine whether the occlusion was successful. Those mice that had a surface cerebral blood value in the ischemic core that was higher than 15% of the baseline or mice that had massive bleeding were removed from the study. All mice were allowed to recover under a warm condition with free access to food and water. The pMCAO sham procedure include a skin cut and craniotomy. The sham mice had received the same amount and duration of anesthesia and analgesia as the mice subjected to pMCAO.

### 4.4. Immunohistochemistry

Three days or 8 weeks after surgeries, after being anesthetized, the mice were sacrificed after being perfused with phosphate-buffered saline to remove blood from the vessels. Brains were collected, frozen in dry ice, and sectioned into 20 μm thick coronal sections using a Crystat (CM1900 Cryostat, Leica, Wetzlar, Germany). Sections were immunostained with primary antibodies specific to CD68 (1:50, AbD Serotec, MCA1957, Raleigh, NC, USA), claudin-5 (1:25, Invitrogen, Carlsbad, CA, USA), CD31 (1:100, Santa Cruz, CA, USA), CD13 (1:500, R&D Systems, Minneapolis, MN, USA), or MBP (1:200, Sigma Aldrich, St. Louis, MO, USA) at 4 °C overnight. The sections were stained with Alexa Fluor 594-conjugated or Alexa Fluor 488-conjugated IgG (1:500, Invitrogen, Carlsbad, CA, USA) after being washed with PBS three times to visualize the positive signals. Negative controls were stained with either the primary or the secondary antibody alone. All slides were covered with Vectashield Hardest Mounting Medium with DAPI (Vector Laboratories Inc., Burlingame, CA, USA). The pictures were taken using a fluorescence microscope (Boprevo BZ-9000, Keyence, Osaka, Japan). All quantifications were performed using NIH Image J v1.63. For quantification of CD68^+^ cells, the number of CD68^+^ cells were counted. For calculation of the ratios of claudin-5 to CD31 and CD13 to CD31, the positively stained area of claudin-5 or CD13 was divided by the positively stained area of CD31.

### 4.5. Statistical Analysis

All quantifications were performed by more than two researchers who were blinded to the experimental groups. Data were presented as mean ± standard deviation (SD). Differences among groups were analyzed by one-way ANOVA followed by multiple comparisons with Tukey’s correction using Graphpad Prism 6 (San Diego, CA, USA). A *p*-value < 0.05 was considered to be significant. The sample size was estimated according to our previously published data [32] and indicated in figure legends.

## 5. Conclusions

BF shortly before stroke enhanced microglia/macrophages accumulation and BBB breakdown in the hippocampal CA1 region, and exacerbated WM damage in the basal ganglia of stroke mice, which could contribute to the previously observed long-lasting memory dysfunction of mice subjected to BF and stroke [32].

## Figures and Tables

**Figure 1 ijms-21-08481-f001:**
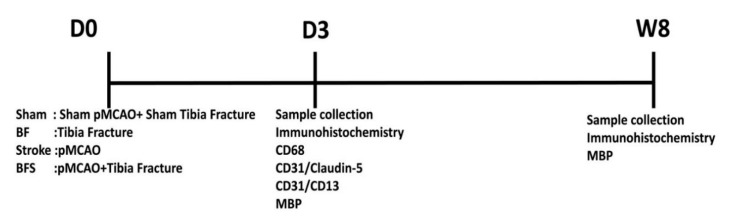
A flow chart illustrating the experimental design. D: day; W: week. CD68 was used as a maker to detect activated microglia and infiltrated macrophages, CD31 for endothelial cells, CD13 for pericytes, claudin-5 for tight-junction protein, and MBP (myelin basic protein) for white matter (WM) measurement.

**Figure 2 ijms-21-08481-f002:**
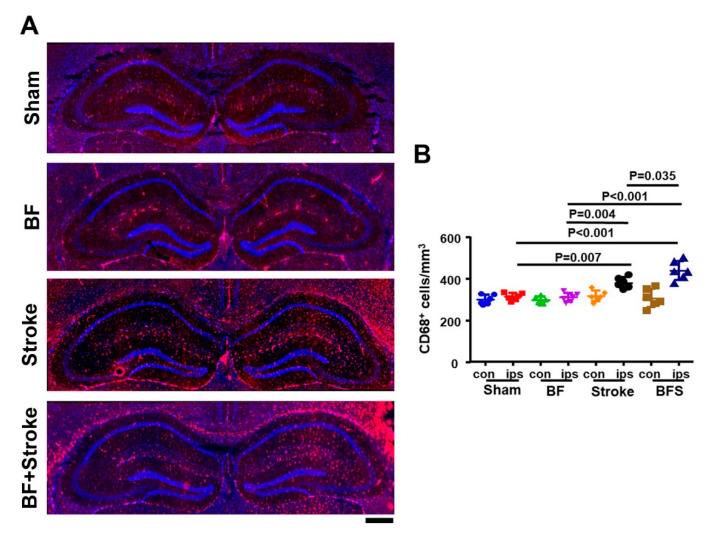
BF increased the amount of CD68^+^ cells in the ipsilateral hippocampal CA1 region. (**A**). Representative microscopic images of CD68^+^ antibody (red) stained sections. The nuclei were stained with DAPI (Blue). Scale bar: 500 µm. (**B**). Quantification of the number of CD68^+^ cells. The groups represented by different colors are indicated on the X-axis. *N* = 6. con: contralateral to stroke injury. ips: ipsilateral to stroke injury. BFS: BF 6 h before stroke.

**Figure 3 ijms-21-08481-f003:**
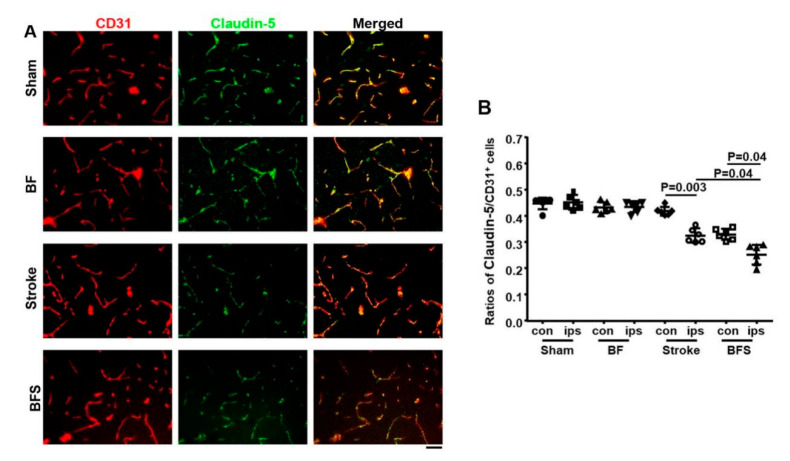
BF enhanced the reduction of claudin-5 protein in the ipsilateral hippocampi of stroke mice. (**A**). Representative images of sections stained with anti-CD31 (red) and anti-claudin-5 (green) antibodies. Scale bar: 50 µm. (**B**). Quantification of claudin-5 and CD31 ratios. *N* = 6. con: contralateral to stroke injury. ips: ipsilateral to stroke injury. BFS: BF 6 h before stroke.

**Figure 4 ijms-21-08481-f004:**
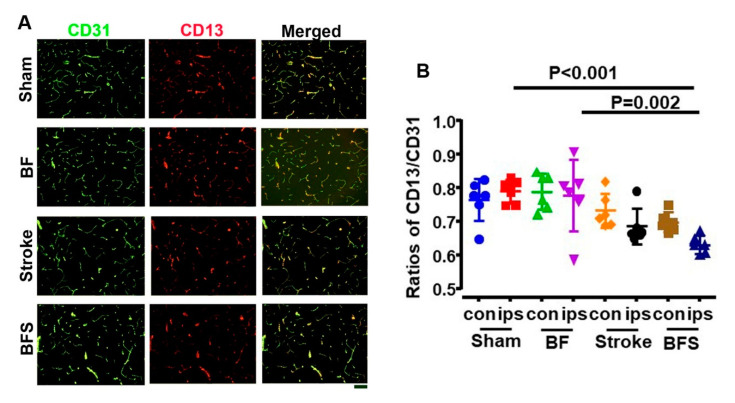
BF reduced vascular pericyte converge in the hippocampi ipsilateral to the stroke injury. (**A**). Representative images of sections stained with antibodies specific to CD31 (green) and CD13 (red). Scale bar: 50 µm. (**B**). Quantification for the ratios of CD13^+^/CD31^+^ cells. The groups represented by different colors are indicated on the X-axis. *N* = 6. con: contralateral to stroke injury. ips: ipsilateral to stroke injury. BFS: BF 6 h before stroke.

**Figure 5 ijms-21-08481-f005:**
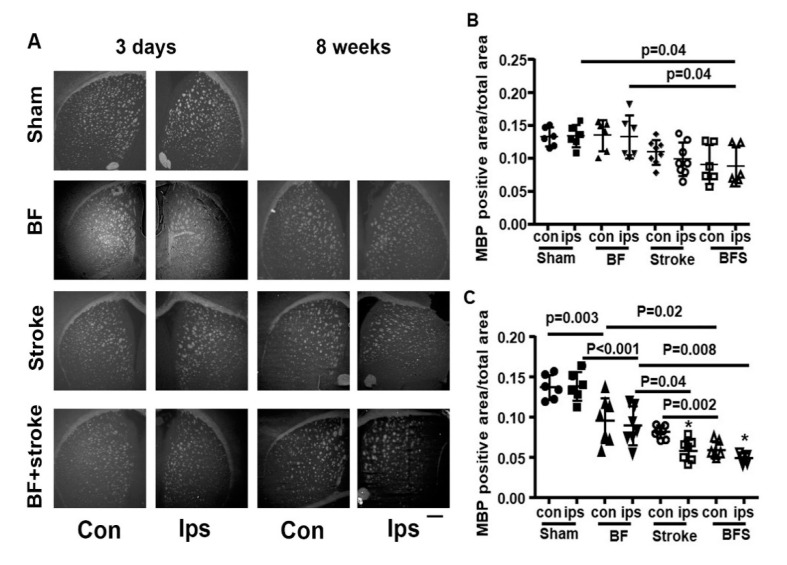
BF exacerbated WM damage in the basal ganglia of the stroke brain. (**A**). Representative images of MBP antibody stained (white) sections. 3 days: 3 days after injuries. 8 weeks: 8 weeks after the injuries. Scale bar: 500 µm. (**B**, **C**). Quantification for MBP positive area in basal ganglia at 3 days (**B**) and 8 weeks (**C**) after the injuries. *: *p* < 0.001 vs sham ipsilateral side of stroke. *N* = 5–8. con: contralateral to stroke injury. ips: ipsilateral to stroke injury. BFS: BF 6 h before stroke.

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
