# Peer review of "Bone Fracture Enhanced Blood-Brain Barrier Breakdown in the Hippocampus and White Matter Damage of Stroke Mice"

_ijms, 2020, doi:10.3390/ijms21228481_

Round 1
Reviewer 1 Report
This study addressed the effects of a tibial fracture prior to ischemia in a mouse stroke model. Overall, the manuscript is well-written with a few editorial errors. There are some important missing information in the introduction and methods and there are a few issues that need to be addressed in the results and discussion. These are outlined below.
Introduction:
Major comments
The authors need to provide some clinical relevance and more information in the background about any cognitive or neurobehavioral changes seen in patients with tibial (or other kinds) of fracture with and without stroke. The authors do provide some background about animal models but what do we see clinically? Do patients who fall and break a leg show memory or other cognitive deficits? This is extremely important for emphasizing the significance of the study and this information is available in the medical literature.
Minor comments
Line 35- “comparing” should be “compared”
Line 54- subject-verb agreement: “lead” should be “leads”
Methods:
Major comments
My main concern with this study is that the sham group for the bone fracture surgery. Why did the authors’ not make a skin incision in the hindlimb of sham animals? Not having an incision in sham group could mean that those animals had less systemic inflammation, which (based on the authors’ hypothesis and findings) may translate into less neuroinflammatIon and BBB disruption. Since assessing inflammation was one of the endpoints of this study, this is a confounding variable that the authors have to address. The authors need to explain why the incision was not made and they must provide a detailed discussion of how this could have affected their data. This is an important limitation of this study.
The authors must also provide how the immunohistochemical quantitative analyses were performed. 1) What software did the authors’ use? 2) Did the authors quantify overall immunofluorescence or did they do cell counts? 3) The authors need to describe that they calculated ratios of claudin to CD31 and so on and why they chose to do that. None of this is clear from the description the authors provide.
Minor comments
Line 189- “through 2% isoflurane” should be “with 2% isoflurane”
Line 189 – delete “the” before buprenorphine
Line 191- delete “the” before lidocaine
Results:
Major comments
The description of the results needs to be expanded and the authors. For example, there needs to be an explanation of what a ratio of claudin5 to CD31 means and what having significant different ratios indicates biologically. The same applies for CD13:CD31. Also, for the WM assessment, the authors should cite some of the area values ± SD that they found, instead of just reporting the p values.
An interesting result that the authors show in the graphs, but do not address at all, is that there is consistently more variability in the BF group than any of the other groups. The authors should, first, describe this finding and, second, discuss what this could mean in the discussion.
Finally, why was WM examined only in the basal ganglia? Did the authors look in other areas and not find a difference? If so then this should be reported in the results. If the authors had a specific reason for examining the basal ganglia, then this needs to be explained and the proper background should be provided in the introduction and the results further discussed n the discussion.
Minor comments
Line 106- that should be than.
Discussion:
Major comments
- The authors should provide some proposed mechanism for how a peripheral bone injury can lead to the central effects the authors describe. This is a big piece that is missing in the discussion as it stands.
- In the third paragraph of the discussion, line 158, the authors mention that pericytes as the major cellular component of the BBB. This is simply wrong. The pericytes are an important part of the BBB, but the BBB is formed (as the authors mention!) by endothelial cells. It is prudent that authors select their words carefully. This needs to be revised.
- Here again the authors need to explain why they looked at WM in the basal ganglia. There also needs to be a discussion of what a reduction in WM in the basal ganglia indicate or be associated with behaviorally.
- The authors need to make a connection between their different findings. Specifically, how are neuroinflammtion, BBB disruption and WM damage connected? Are there other studies that show that microgliosis for instance is associated with injury to the endothelium and BBB disruption (or vice versa) and WM damage? (There are and the authors need to incorporate some of these studies into the discussion).
- Finally, there are limitations to the conclusions drawn from this study, which the authors need to address.
Minor comments
Line 152- delete the word “forgetting”
Line 153-154 please revise this sentence. Something is missing.
Reviewer 2 Report
This Scientific research does not provide scientific value or novelty and, in my opinion, is not eligibile for printing.
- The researchers have not provided the flow chart of the study, the research methodology was not fully understood, and why was such a methodology – BF, stroke, BF+ stroke?
- The interaction between bone fracture and stroke and its influence on long-lasting memory dysfunction is not understandable, especially the direct interaction between stroke and bone fracture, see cited scinetific literature: a) Post-stroke bone fracture negatively impacts stroke recovery, prolongs hospital stays, and increases economic cost. Stroke, osteoporosis and bone fracture share common risk factors. The main risk factors for post- stroke bone fracture include aging, osteoporosis, and loss of posture control. Bone mineral density measurement may identify patients who are at risk of post-stroke bone fracture [5], b) The increased risk of fractures following stroke is hypothesized to be related to both poststroke declines in bone mineral density and to an increased risk of falls [3].
- Analysis of statistical calculations of the survey samples is not provided, a broader data context is required
- The authors emphasize that all animal testing procedures have been approved by the Institutional Animal Care, but no protocol number is provided, Year?
- What kind of mice, their main characteristics?
- The findings of the study do not support the scientific hypothesis.
- We would like to have a more detailed and in-depth analysis of the scientific results obtained, comparing it with other researchers
Reviewer 3 Report
This manuscript investigated the mechanism of tibia fracture (BF) shortly before stroke causes long-lasting memory dysfunction in mice. They hypothesis that BF enhances neuroinflammation and blood brain barrier (BBB) breakdown in the hippocampus and white matter (WM) damage which may contribute to the long-lasting memory dysfunction in BF+stroke mice. This is a interesting study. However, there are few concerns need to be addressed.
- In the results, please proved stroke group Claudin-5 immunostaining figure in Figure 2.
- In the results, please provide MBP immunostaining pictures in Figure 3.
- In the discussion, authors should discuss Please add limitation for possible mechanisms of how tibia fracture increases brain neuroinflammation. How a system injury interacts with brain and increases neuroinflammation and induces worse brain damage.
Round 2
Reviewer 2 Report
Thanks for the corrections. I have no substantive comments.